# SWATH Based Quantitative Proteomics Reveals Significant Lipid Metabolism in Early Myopic Guinea Pig Retina

**DOI:** 10.3390/ijms22094721

**Published:** 2021-04-29

**Authors:** Jingfang Bian, Ying-Hon Sze, Dennis Yan-Yin Tse, Chi-Ho To, Sally A. McFadden, Carly Siu-Yin Lam, King-Kit Li, Thomas Chuen Lam

**Affiliations:** 1Centre for Myopia Research, School of Optometry, The Hong Kong Polytechnic University, Kowloon, Hong Kong, China; bian.jf.bian@connect.polyu.hk (J.B.); 18074319r@connect.polyu.hk (Y.-H.S.); dennis.tse@polyu.edu.hk (D.Y.-Y.T.); chi-ho.to@polyu.edu.hk (C.-H.T.); carly.lam@polyu.edu.hk (C.S.-Y.L.); kk.li@polyu.edu.hk (K.-K.L.); 2Centre for Eye and Vision Research (CEVR), 17W Hong Kong Science Park, Hong Kong, China; 3School of Psychology, College of Engineering, Science and the Environment, University of Newcastle, Callaghan, NSW 2308, Australia; 4Shenzhen Research Institute, The Hong Kong Polytechnic University, Shenzhen 518052, China

**Keywords:** SWATH-MS, proteomics, myopia, retina, guinea pigs, lipid metabolism

## Abstract

Most of the previous myopic animal studies employed a single-candidate approach and lower resolution proteomics approaches that were difficult to detect minor changes, and generated limited systems-wide biological information. Hence, a complete picture of molecular events in the retina involving myopic development is lacking. Here, to investigate comprehensive retinal protein alternations and underlying molecular events in the early myopic stage, we performed a data-independent Sequential Window Acquisition of all Theoretical Mass Spectra (SWATH) based proteomic analysis coupled with different bioinformatics tools in pigmented guinea pigs after 4-day lens-induced myopia (LIM). Myopic eyes compared to untreated contralateral control eyes caused significant changes in refractive error and choroid thickness (*p* < 0.05, *n* = 5). Relative elongation of axial length and the vitreous chamber depth were also observed. Using pooled samples from all individuals (*n* = 10) to build a species-specific retinal ion library for SWATH analysis, 3202 non-redundant proteins (with 24,616 peptides) were identified at 1% global FDR. For quantitative analysis, the 10 individual retinal samples (5 pairs) were analyzed using a high resolution Triple-TOF 6600 mass spectrometry (MS) with technical replicates. In total, 37 up-regulated and 21 down-regulated proteins were found significantly changed after LIM treatment (log2 ratio (T/C) > 0.26 or < −0.26; *p* ≤ 0.05). Data are accepted via ProteomeXchange with identifier PXD025003. Through Ingenuity Pathways Analysis (IPA), “lipid metabolism” was found as the top function associated with the differentially expressed proteins. Based on the protein abundance and peptide sequences, expression patterns of two regulated proteins (*SLC6A6* and *PTGES2*) identified in this pathway were further successfully validated with high confidence (*p* < 0.05) using a novel Multiple Reaction Monitoring (MRM) assay on a QTRAP 6500+ MS. In summary, through an integrated discovery and targeted proteomic approach, this study serves as the first report to detect and confirm novel retinal protein changes and significant biological functions in the early LIM mammalian guinea pigs. The study provides new workflow and insights for further research to myopia control.

## 1. Introduction

Myopia is a common refractive error worldwide and the uncorrected refractive error ranks as the second leading cause of irreversible blindness globally [1]. In east Asian countries, over 80% of teenagers and young adults are myopic which demonstrates early onset epidemiologic features [2]. It is accepted as a multifactorial disorder and characterized by parallel light focused in front of the retina rather than on the retina of a relaxed eye [3]. Due to a myopia boom worldwide, the next challenge is the increased percentage of high myopia [4,5]. The increased prevalence of high myopia will increase risks of cataract [6,7], glaucoma [8,9], and retinal detachment [10,11]. Although various clinical approaches have been designed to correct myopia, the underlying molecular mechanisms controlling ocular growth during myopia progression are still unclear.

Various species of animals have been introduced and involved in experimental myopic research, such as chicks [12,13], mice [14,15], tree shrews [16,17,18], guinea pigs [19,20,21], and monkeys [22,23,24,25]. Among them, the guinea pig (*Cavia porcellus*) provides a popular mammalian model that has increased in use as an animal model for studying ocular growth in recent years [19,20,26,27]. The biometric changes in developing guinea pig eyes are more similar to those in monkeys compared to common widely used myopic chicken models under similar experimental conditions [20]. The key aspects of retinal development in guinea pigs were also similar to humans [28]. In addition, it was also easy to perform ocular parameters’ measurements, obtain large ocular tissues and induce myopic changes reliably and reproducibly by optic interventions [20,27].

Within the visual pathway, the retina is the first place to detect light signals and convert the light signals to neuronal signals. In myopia studies to date, it has been widely accepted that the retina is the primary site to detect a defocus signal, and transmits biochemical signals through the choroid to cause tissue remodeling in the sclera [13,29]. These biochemical signals cause eyeball elongation, resulting in myopia. Previous studies have also shown that signals responsive to defocus during ocular development originate from the retina and no higher center within the CNS is required [30,31]. Moreover, the expression level of numerous genes in the retina was also changed after lens induced myopia (LIM) and form-deprivation myopia (FDM) treatments [29].

As a traditional ‘top-down’ proteomics technique, 2-DE gel based proteomics approaches played an important role to give insights into physiology of diseases and applied in various tissues [32,33]. Although some advantages of it exist, such as identifying PTM and protein isoforms, there are some limitations of gel based proteomics, such as dynamic range, difficulties with protein fractionation of specific spots in gel, protein ionization and fragmentation [34]. Therefore, the throughput and sensitivity of gel-based proteomics is limited. ‘Bottom-up’ technique, also called shotgun proteomics, has rapidly expanded to provide a new direction of gel free quantitative proteomics [35,36]. It supplies an indirect measurement to proteins by proteolysis peptides from protein mixtures with obvious advantages of high sensitivity, throughput, and accuracy [37].

In proteomic studies of myopia, some earlier studies based on 2-D PAGE and 2-D DIGE proteomics have been adopted, with some regulated proteins revealed in myopic animal models [38,39,40,41]. Recently, data-independent acquisition (DIA) in mass spectrometry has proven to be a powerful proteomic approach to study complex biological samples with high reproducibility, speed, and resolution [42,43], and was applied in the TripleTOF system (SCIEX), also known as sequential window acquisition of all theoretical mass spectra (SWATH) [44]. In ocular research, the SWATH-MS based proteomic approach has been successfully used in various ocular diseases and conditions, such as the retinal proteome during normal growth [45], human tear [46], glaucoma [47], and oxygen-induced retinopathy [48]. This platform has proven to be powerful for comparative analysis of protein expression or regulation among different tissues in eye research. However, no publication investigates the differentially expressed proteins and underlying molecular events in retina involving early myopic development in the mammalian guinea pig model. The recent development of the multiple-reaction monitoring (MRM) technique can quantify multiple proteins of interest in a single experiment without antibody [44,49] with the advantages of higher throughput and high accuracy [50]. It is a well-established method to quantify small molecules [51,52], and gradually applied in protein biomarker verification in human diseases [53,54]. Therefore, apart from SWATH based discovery proteomic approach, MRM-MS was further selected to validate selected regulated proteins involved in lipid metabolism. In brief, the main aims of this work were to characterize the whole retinal proteome, differentially expressed proteins and potential biochemical pathways at the early stage of myopia using a novel SWATH-based quantitative proteomics. In addition, the index proteins associated with lipid metabolism were also validated using MRM-based targeted proteomics, which could supply new insights for the mechanisms underlying myopia, and this integrated proteomic workflow also provides a novel molecular protocol for other eye research.

## 2. Results

### 2.1. Ocular Parameters Changes

LIM was initiated in 5 pigmented guinea pigs when they were 4 days old in one eye. After 5D lens wear for 4 days, significant relative myopia was observed (myopic vs. control: 5.25 ± 2.33D vs. 6.94 ± 1.81D, difference: −1.69 ± 0.65D, mean ± SD, *p* = 0.004). In addition, a significant and relatively thinner (−0.017 ± 0.010 mm) choroid thickness (myopic vs. control: 0.16 ± 0.01 mm vs. 0.17 ± 0.02 mm, mean ± SD, *p* = 0.016) was found in myopic eyes. Moreover, although the difference did not reach statistical significance, relative elongation (0.018 ± 0.040 mm) in axial length was observed in myopic eyes (myopic vs. control: 7.416 ± 0.084 mm vs. 7.397 ± 0.104 mm, mean ± SD, *p* = 0.361). There was a correspondingly relative enlargement (0.024 ± 0.041 mm) in vitreous chamber depth when myopic eyes were compared to control eyes (myopic vs. control: 2.98 ± 0.03 mm vs. 2.96 ± 0.02 mm, mean ± SD, *p* = 0.253). Other ocular parameters showed no significant difference after 4 days monocular LIM treatment (Table 1).

### 2.2. Generation of the Ion Library Using a Pooled Retinal Proteome

After 4-day LIM, according to the protein assay, no significant difference was found in terms of the total protein concentrations between the myopic and control eyes using paired Student’s *t*-test. Approximately 3 μg retinal peptides from all the myopic and control eyes of 10 animals were used to generate an ion library. After meeting identification criteria mentioned in the method (at global 1% FDR), a total of 3,202 unique, non-redundant proteins and 24,616 peptides (Appendix A for a full list of identified proteins; Appendix A for protein and peptide FDR analyses, respectively) were generated. It also served as the ion library to support quantitative analysis in SWATH acquisition. In addition, STRING has proven to be a powerful database to give some information of known interactions and predict interactions among proteins [55]. Among all identified proteins, 2507 gene names were mapped, the top 2000 IDs were uploaded due to the limited upload size to STRING to check interactions among them. In total, 1947 nodes (gene names) and 57,986 edges (predicted functional associations) were observed (Appendix A).

### 2.3. Gene Ontology Analysis of a Pooled Retinal Proteome

Functional analysis of all identified retinal proteins was performed by the PANTHER gene classification analysis. According to gene ontology (GO), 2507 proteins (~80%) were mapped to relevant homo sapiens gene names. The top three molecular functions of retinal proteins were “catalytic activity” (42.10%), “binding” (37.20%), and “transporter activity” (7.10%) Figure 1A), and the top three biological processes were “cellular process” (31.56%), “metabolic process” (25.65%), and “cellular component organization” (10.42%) (Figure 1B). In the cellular component, “cell part” (40.14%), “organelle” (26.43%), and “macromolecular complex (18.52%) (Figure 1C) were identified as the major three groups in current retinal samples.

### 2.4. Data Quality Check of All SWATH Dataset

The coefficient of variation (CV) was performed to evaluate reproducibility and variability in all the SWATH injections (10 biological samples, 3 technical replicates, 30 MS injections in total) matched to IDA library by SWATH^®^ Replicates Analysis 2.0 Toolkit (SCIEX) [56]. In total, 2901 proteins and 17,812 peptides were matched among all samples after SWATH processing. Finally, about 86.6% and 75.3% of proteins and peptides, respectively, were found to have less than 30% CV across multiples runs (30 injections) using VW100 acquisition in SWATH (Figure 2). All the raw data generated from IDA and SWATH acquisitions has been submitted and published in the PRIDE [57] repository for general release (PXD025003).

### 2.5. Pathways Analysis of All Significantly Expressed Proteins

IPA software was used to investigate potentially important pathways involved in early myopia development after 4-day LIM. All the mapped 2797 genes names (87% of identified proteins) used for quantitation were uploaded into IPA online-based bioinformatics application. Processed filtering defaulted to *p* < 0.05 as described in previous studies [58,59]. The module of the IPA canonical pathways displayed the most significant canonical pathways based on the *p* values in the entire dataset. The *p* values for the canonical pathways were calculated by two ways: (a) the ratio of the number of molecules from the current dataset that map to the pathway divided by the total number of molecules that map to the canonical pathways; and (b) *p* values obtained from Fisher’s exact test (right-tailed) to indicate the probability of random association of molecules from our dataset with the canonical. Statistically significant pathways were defined as *p* < 0.05. Finally, 90 significant pathways were predicted by the pathway analysis module (Table 2). Then, 25 significant pathways could crosstalk with each other by shared common genes (Appendix A). Based on the results from IPA pathway analysis, 56 of them were found matched to the significant pathways in previous reports at gene level or protein expression levels. (Table 2). The top one significant pathway was phototransduction pathway.

### 2.6. Quantitative Analysis of Differentially Expressed Proteins

After individual injection of 10 retinal samples with 3 technical replicates, 30 SWATH injections were analyzed at the same time. Differentially expressed proteins between the treated and control eyes were quantified and compared using the same ion library after retention time alignment and normalization. In summary, 37 proteins were found to be significantly up-regulated, and 21 proteins were down-regulated when comparing the myopic eyes to the control eyes (log2 ratio (T/C) > 0.26 or < −0.26; *p* ≤ 0.05). The ggplot2 package in the R programming language was selected to generate a volcano plot, which showed the differentially expressed proteins (Figure 3). The fold change was converted to log2 fold change in the *x*-axis, and the *p* value was converted to –log10 *p* value in the *y*-axis. The detail of differentially expressed proteins are shown in Appendix A.

In addition, 58 differentially expressed were also uploaded to STRING to check interactions among them. 56 nodes (gene name) and 20 edges (predicted functional associations) were observed (Figure 4). The top one enriched pathway was metabolic pathways (has0100) based on KEGG pathways database with 0.0423 false discovery rate. In total, 11 proteins were involved in metabolic pathways and labeled with red dots in Figure 4.

For these significantly regulated proteins, ‘disease and function analysis’ was also performed using Ingenuity Pathway Analysis (IPA). Significance (−log (*p* value)) for the GO module ‘disease and function analysis’ was calculated by two factors: (1) the number of functional analysis molecules involved in that particular function; and (2) the total number of molecules which were associated with that function in the Ingenuity Knowledge Base. The right-tailed Fisher’s exact test was used to calculate the *p* value determining the probability that each biological function assigned to the dataset was due to chance alone. Using this module, “lipid metabolism” was finally identified as the top function associated with the differentially expressed proteins (Figure 5A). The 15 differentially expressed proteins that are associated with the GO pathway of “lipid metabolism” are shown in (Figure 5B), including 12 up-regulated proteins (80%) and 3 down-regulated proteins (20%).

### 2.7. Validation of Index Proteins in Lipid Metabolism Using MRM Based Target Proteomic Approach

The 15 regulated proteins involved in lipid metabolism were selected for further validation using the MRM based targeted proteomic approach. To eliminate sample variation because of biological replicates and sample preparation, the top one transition (AITIFQER_+2y6, ratio (T/C) = 0.9) with the largest peak area of Glyceraldehyde 3-phosphate dehydrogenase (*GAPDH*) protein was selected as the internal standard transition. Then each transition peak area was normalized by matched transition of *GAPDH* in the same biological sample. Overall, 8 regulated proteins were removed due to low abundance during optimizing the MRM method. Finally, 8 proteins (12 peptides with 36 transitions) were left for the final MRM experiment (Appendix A).

Upregulation of *SLC6A6* and *PTGES2*, both involved in lipid metabolism, were confirmed with the most evidence of differential expression in the 4-day LIM group, where it met Level 2 confidence criteria. In addition, upregulation of other 5 proteins (*CP*, *CDS1*, *SPTLC1*, *CCDC22*, and *GLA*) also met Level 1 confidence criteria in the 4-day LIM group (Table 3).

## 3. Discussion

### 3.1. Changes in Refractive Error and Ocular Dimension

In previous studies, a clear myopic shift in terms of refractive error was found after treating guinea pigs with 4D lens for eight days [40] and 5D lens for 11 days [83]. A relative myopic shift was also found in treated eyes after 4-day LIM, which validated the myopia model and was consistent with the findings reported in previous studies. Although no statistically significant difference in ocular dimension was found, the myopic shift of VCD and AL in treated eyes indicates that the changes were the result of lens inducement, not by chance. However, a previous study also showed statistically significant differences in VCD and AL after inducing myopia for eight days using 4D lens [40]. This difference in statistical outcomes is likely due to the shorter duration of myopia induction in current study. The choroid thickness also showed a tendency of thinning in the treated eyes, which was consistent with the findings of a previous study in myopic guinea pigs after 10 days of myopia induction using 4D lenses [20]. Moreover, AL only accounts for approximately 50% change in refraction in human eyes [84]. Corneal radius of curvature ratio also induces the changes in refractive error [85]. However, in the current study data for the corneal radius of curvature ratio were not available. In summary, the changes in refractive error, VCD, AL, and choroid thickness suggested that the eye was at an early stage of myopia development.

### 3.2. Protein Ion Library

The total number of proteins identified in the current study was similar to the total number of proteins identified in one recent publication using guinea pig retina during emmetropization [86], suggesting that this SWATH-based proteomic method performs consistently when applied in multiple retina research paradigms. The results of the GO analysis for categorizing identified proteins according to their biological process and molecular function in the current study was consistent with a previous study, which reported that the highest GO categories in three distinct human retinas were “cellular process”, “binding”, “catalytic activity”, and “intracellular regions” [87]. Similar protein categories of retinal proteins were also found when comparing guinea pigs to chicks [88,89], suggesting that retina tissues shared similar GO categories between these two popular myopia animal models and human retina.

### 3.3. Significant Pathways

In the UniProt database, the full annotation of proteins for guinea pigs is not completed; all differentially expressed proteins were converted to corresponding gene annotations and pathway analysis performed. Finally, 90 significant pathways in total were found. Among them, our single experiment using SWATH-based proteomics results generated 56 significant pathways, which had also been reported as regulated in myopia across multiple previous studies either at the gene or protein level (Table 2), suggesting that this SWATH-MS based proteomic approach is a powerful technique to yield more relevant pathways in a single experiment than other techniques. The result also validated the SWATH method, because these identified pathways have been corroborated by other studies using conventional analyses, such as quantitative PCR, Western blotting, and MRM.

Among all the identified pathways, the top number one pathway was the ‘phototransduction pathway’. Phototransduction refers to a process by which light is converted into electrical signals in the rod cells and cone cells in the retina [90]. The critical role of phototransduction was proposed in two recent eye growth studies involving the retina. It was found to be a significant pathway implicated in myopic marmoset retinal gene expression in response to optical defocus of opposite signs [61] and guinea pig retina during the normal emmetropization process we recently published using the same SWATH-MS technique [45]. In the current study, 7 significant proteins were found involved in this pathway, in which two proteins were reported in the previous studies. Specifically, Phosphodiesterase-6 (PDE6), as the key effector enzyme of phototransduction cascade, its mutation could result in hyperopia and axial length shortening in the rd1 and rd10 mice, compared with the wild-type [91]. Guanine nucleotide-binding protein (*GNB1*) was known as a modulator or transducer in various transmembrane signaling systems. One previous study suggested it could contribute to a 50% reduction in rod transduction and retinal degeneration in the Rd4 mice [92]. Any other 5 candidates involved in phototransduction pathways were not well-studied in other eye diseases, such as *PRKACB*, *RGS9BP*, *GRK1*, *PRKAR1A*, and *RCVRN*. Interestingly, another well-established pathway involved in myopia, ‘dopamine receptor signaling’ was also found in this study. Extensive studies have reported dopamine (DA) as a “stop” signal in ocular growth [74,76]. A non-selective DA receptor agonist (apomorphine) prevents form deprivation myopia (FDM) in several animal models, including in chickens [93,94], guinea pigs [95], monkeys [96], and mice [97]. Serine/threonine-protein phosphatase 2 (*PPP2R2*) is also involved in this pathway. *PPP2R2* is a group of enzymes that catalyze the removal of phosphate groups from serine and threonine residues by hydrolysis of phosphoric acid monoesters. They oppose the action of kinases and phosphorylases and are involved in signal transduction, but have never been previously reported as changing in myopic tissues. In a previous study, *PPP2R2* has been reported to be involved in the MAPK signaling pathway [98], and this pathway has a close relationship with myopia based on a clinical gene analysis [99] and a basic animal study [100]. Hence, our SWATH-based proteomic approach may yield more comprehensive proteins, including novel candidates, to the established pathways in myopia research. In addition, SWATH-MS based proteomics may also help us to find novel pathways involved in myopia progression, such as the Hippo pathway, which was recently suggested to have a vital role in myopic marmosets [61] and mice [60] at the gene expression level. However, regulation of the Hippo pathway in myopia models has not been demonstrated at the protein level. In summary, different from the conventional tools of closed target methods [42,101], the overall results in the current study suggest that the SWATH-based proteomic method used was a large-scale open target method.

Using IPA analysis, 25 significant pathways were found interacted, which supplied detailed pathway crosstalk maps which may signify the molecular events underlying the early stage of myopia development (Appendix A). Significantly expressed proteins were found associated with several significant pathways, such as phototransduction pathway, oxidative phosphorylation, mitochondrial dysfunction, IL-1 signaling, and melatonin signaling. Although many of these pathways were reported to be involved in previous myopia studies from a number of scattered and unrelated studies, the current dataset helps uncover network-associated pathways in a single system combining SWATH based proteomic approach and bioinformatics analysis.

### 3.4. Index Proteins Invovled in Lipid Metabolism

Through SWATH data quality evaluation using CV, we found roughly 80% of the peptides and proteins had a CV of less than 30%, which was similar to the previous study using SWATH based quantitative proteomics approach [102]. After a stringent criterion (log2 ratio (T/C) > 0.26 or < −0.26; *p* ≤ 0.05) was applied for identification of differentially expressed protein, 58 regulated proteins were found, which also provided the largest set of differentially expressed proteins identified in a single study of early myopia.

To filter proteins of interest involved in the most significant molecular function, we performed functional analysis of the 58 significant regulated proteins. Lipid metabolism was found as the top number one most significant function in guinea pig retina during early myopic development. Previous studies have also reported the important role of lipid metabolism in myopia [103,104,105], and mainly based on single target study or metabolism study, different to the current approach. To further confirm the important role of lipid metabolism, an additional validation of SWATH results was accomplished using the MRM based proteomic approach, which has become a new method for targeted quantitation in a typical proteomics pipeline due to its prominent advantages of strong specificity, high sensitivity, high accuracy, and good reproducibility [106,107]. After validation study, *SLC6A6* and *PTGES2* were successfully validated with the most evidence, and another 5 differentially expression proteins involved in lipid metabolism were also confirmed with the same direction in differential expression by both SWATH based discovery and the MRM based target proteomic approach. This indicated that these two methods were complementary (Table 3). The finding also supported the significant role of lipid metabolism at the early stage of myopia, with 7 notable associated proteins up-regulated, as briefly discussed below.

Sodium- and chloride-dependent taurine transporter 6 (*SLC6A6*) can transport taurine and beta-alanine. Gene ontology (GO) annotations of this gene include neurotransmitter of sodium symporter activity and taurine transmembrane transporter activity. *SLC6A6* has never been previously reported as involved in myopia pathways. However, in ocular studies, one previous study found that biallelic mutation of human *SLC6A6* could cause early retinal degeneration [108]. In addition, Sodium- and chloride-dependent taurine transporter 6 belong to the family of Na+- and Cl−-dependent neurotransmitter transporters, and is also known as one of *GABA* transporters [109]. One recent retinal study using rats found *SLC6A6* had the ability to use GABA as a substrate using *SLC6A6*/TauT-overexpressing cells and *SLC6A6*/TauT-mediated GABA transport was present at the inner blood-retinal barrier (BRB) using a conditionally immortalized rat retinal capillary endothelial cell line [110]. In previous myopia research, an increased retinal GABA concentration and increased retinal mRNA expression of type-A and type-C GABA receptors were found in guinea pig LIM [111]. Interestingly, a recent study in guinea pigs has shown that LIM also leads to increased expression of GABA and GABA receptors in the visual cortex, similar the LIM-mediated increased expression of the retinal GABA [112]. Therefore, the increased level of *SLC6A6* may prompt myopia by using GABA as a substrate.

Prostaglandin E synthase 2 (*PTGES2*) is an isomerase that catalyzes the conversion of *PGH2* into the more stable prostaglandin E2 (*PGE2*). There is very limited previous study of the involvement of this protein in myopia research. In glaucoma studies, prostaglandin analogues are the front-line medications for the treatment of glaucoma [113]. The down-regulated of *PTGES* was also found in gene expression level compared glaucoma samples to normal samples [114]. In myopia research, topical latanoprost, as a representative prostaglandin analog, was effective for lowering IOP and slowing myopia progression in FDM guinea pigs [115]. However, all these studies did not explain underlying mechanisms. Therefore, the up-regulated of *PTGES2* in LIM guinea pig retina was opposite to the prevention effect of prostaglandin analog in FDM guinea pigs. It may be explained by different mechanisms underlying LIM and FDM.

Ceruloplasmin (*CP*) belongs to an α-2 glycoprotein and is associated with the transportation of copper in the bloodstream [116]. Ceruloplasmin cannot cross the blood-brain and blood-retinal barriers and is present in the central nervous system (CNS) and synthesized locally [117]. In previous glaucoma studies, by RT-PCR and immunoblotting analysis, *CP* mRNA and *CP* protein were up-regulated in the retinas of glaucomatous DBA/2 mice, which indicated triggering by the IOP stress and then lead to retinal ganglion cell (RGC) loss in the retina. Moreover, up-regulation of *CP* was also found in most human eyes with glaucoma and localized to the Muller cells within the retina and in the area of the inner limiting membrane [118]. In previous myopia research, only one study mentioned 90–95% of serum Cu was related to ceruloplasmin, but there were no significant differences in serum Cu concentration in myopic children in comparison to the control group [119]. Taken together, the up regulation of ceruloplasmin protein levels involved in early myopia found here for the first time may be triggered by initial ocular enlargement stress similar to findings in glaucoma.

Phosphatidate cytidylyltransferase 1 (*CDS1*), also known as CDP-Diacylglycerol Synthase 1, is an enzyme utilizing phosphatidic acid (PA) to produce CDP-DAG, which is an important precursor for the synthesis of phosphatidylinositol, phosphatidylglycerol, and cardiolipin. The overexpression of this protein may amplify cellular signaling responses from cytokines and plays an important role in retinal signal transduction. *CDS1* has never been previously reported as a factor involved in myopia development. However, in zebrafish embryos, *CDS1* mutation could reduce VEGFA signaling activity and cause an inefficient angiogenesis phenotype [120]. In ocular studies, one retinal study found down-regulated *CDS1* in retinal degeneration rd3 mouse by RNA microarray analysis [121]. One previous myopia study also found Bevacizumab could inhibit FDM chicks by about 50% through binding to VEGFA [122]. Therefore, the increased level of Phosphatidate cytidylyltransferase 1 may prompt myopia via enhancing VEGFA signaling activity.

Coiled-coil domain-containing protein 22 (*CCDC22*) is involved in the regulation of nuclear factor kappa-light-chain-enhancer of activated B cells (NF-kB) by interacting with copper metabolism Murr1 domain-containing (*COMMD*) proteins. There has been no report involving this protein in previous myopia studies. This gene was regarded as a novel candidate gene for syndromic X-linked intellectual disability with ocular manifestations, such as optic atrophy, nystagmus, and strabismus [123]. In addition, *CCDC22* deficiency regulated the inflammatory response by participating in the activation of NF-kB in humans and was associated with *COMMD8*, which is a master regulator in the immune system [124]. Moreover, the expression level of receptor activator of NF-kB in aqueous humor was statistically significantly higher in high myopic patients with cataracts [125]. Because our LIM guinea pig was at the early stage of myopia, the finding may be different from high myopia. Therefore, the increased level of coiled-coil domain-containing protein 22 may inhibit NF-kB at an early stage of myopia. Serine palmitoyltransferase 1 (*SPTLC1*) is the key enzyme in sphingolipid biosynthesis. A previous study found mutations in this gene in patients with hereditary sensory and autonomic neuropathy type I (*HSANI*), which is characterized by prominent sensory impairment [126], but has never been previously reported as involved in myopia. Alpha-galactosidase A (*GLA*) is localized on the X chromosome. This enzyme predominantly hydrolyzes ceramide trihexoside, and it can catalyze the hydrolysis of melibiose into galactose and glucose. This protein was never reported in previous ocular and myopia studies. Mutations of *GLA* are mainly associated with Fabry disease, an X-linked disease [127]. Ocular findings were common and related with disease severity using the Fabry outcome survey Mainz severity score index [128]. Among them, the most vision-threatening complication was retinal vascular occlusion, which could lead to sudden blindness and neovascularization [129]. However, the mutation was different from protein expression. Therefore, more study of this protein is needed.

Despite more differentially expressed proteins being found in the current study using the SWATH based proteomic approach, less common regulated proteins were uncovered compared to previous myopia studies. This disparity may be explained by different animal species (chick, guinea pig, and mice), difficulty of age matching among different animal models and experiments, and sample preparations using different technical approaches (gel-based vs. gel-free). Moreover, although genome assembly of guinea pigs was recently completed with a higher resolution (7×) [130], the total number of fully annotated proteins in guinea pigs is still very limited (organism ID: 10141 in the UniProt database). As of April 2021, only 305 proteins were reviewed out of 25,610 proteins in the proteome.

## 4. Materials and Methods

### 4.1. Animals

Guinea pigs *(Carvia porellus*, Tri-Colored, Wuhan Wanqian Jiaxing Bio-Technology Co., Ltd., Hubei, China) were reared with their mothers in their natural litters and housed in plastic boxes (65 × 45 × 20 cm^3^) in temperature and humidity-controlled rooms. Lighting was provided by overhead white LEDs (luminance of ~500 lux in the box center) on a 12 h light and 12 h dark cycle. Food and water were supplied ad libitum. The housing environment and ocular parameters measurements were similar to our previous studies [45,131]. All experimental procedures include animal breeding, biometric measurements, and tissue collection complied with the ARVO statement. All the animals used in this study were approved by the University of Newcastle in accordance with Australian animal care and ethics legislative requirements. The animals were treated with a minus 5D lens worn in front of one eye from 4 days of age for 4 days and with no lens attached in front of the fellow eye which was used as the control eyes. Animals were euthanized in approximately the middle of the light cycle, and retina was harvested from five animals after 4 days lens inducement (*n* = 5 retina in myopic group and control group, respectively. Retina sample ID number: myopic group; 52R, 54R, 62R, 63R, and 64R; control group; 52L, 54L, 62L, 63L, 64L).

### 4.2. Refractive Error and Ocular Components Measurements

Refractive error was measured by streak retinoscopy (halogen, Welch Allen) in awake animals, which have been treated with 1% cyclopentolate ~1 h before measurement. Guinea pigs refracted with cycloplegia became more hyperopic compared to without cycloplegia [19]. The equivalent sphere was calculated by the mean of powers in the horizontal and vertical meridians as in other studies using the guinea pig model [27]. The high frequency A-scan ultrasonography (Olympus Panametrics Pulser-Receiver 5072PR, Waltham, MA, USA. 18 MHz probe) was used to record ocular component changes. The depth of the anterior chamber and vitreous chamber, the thickness of crystalline lens, retina and choroid were recorded to further analysis. The definition of axial length (AL) was the distance from the anterior cornea to the back of the retina, as same as previously described [19].

### 4.3. Retinal Harvest and Protein Extraction

After 4-days of LIM, guinea pigs were sacrificed by barbiturate overdose, as previously reported [45]. The eyes were enucleated from orbital cavity within 5 min. After shaving extraocular muscles, the eyeballs were hemisected equatorially. The anterior section was discarded. The retina was harvested by isolating from the posterior layers without any retinal epithelium attached. Similar sizes of retina obtained from paired eyes were regarded as successful dissection. Retinal tissues were then transferred to liquid nitrogen and stored at −80 °C until used.

The protein extraction procedure was similar to our previous reports [38,39,40,41,45]. The customized lysis buffer was composed of 7 M urea, 2 M thiourea, 30 mM tris, 2% CHAPS, and 1% ASB14 and protease inhibitor). All retinal tissues are homogenized by a Micro-dismembrator unit (Mikrodismembrator; Braun Biotech, Germany) for 7 min. A tungsten carbide (9 mm) grinding ball inside the chamber was used to help shatter tissues into powder. The retinal tissues coupled with Teflon chamber (3 mL) were cooled down in liquid nitrogen and then homogenized in the dismembrator at 1600 rpm with 100μL of lysis buffer for 6 min. Another 100μL lysis buffer was added into mixture for 1 min more homogenization at 1600 rpm. The samples were kept 4 °C for 20 min. Lastly, all samples were centrifuged at 16,000× *g* for 30 min at 4 °C. The supernatant was collected and stored at −80 °C for protein concentration measurement. The total protein concentration was measured by Bio-Rad Protein assay (Bio-Rad Laboratories, Hercules, CA, USA) according to manufacture.

### 4.4. Individual Retinal Samples Digestion for LC-MS/MS

Then, five paired retinal lysates were prepared from five individual pigmented guinea pigs (myopic group; 52R, 54R, 62R, 63R, and 64R; control group; 52L, 54L, 62L, 63L, 64L). A total of 75 µg of proteins from each sample was reduced in 10 mM DTT (Dithiothreitol) at 37 °C for 1 h. Once finished, samples were alkylated in 40 mM iodoacetamide at room temperature for 45 min in darkness. Then, samples were diluted with 4 x volume of cold acetone coupled with simple vortexing. Incubation procedure was always performed at −20 °C overnight. After this, they were centrifuged at 16,000× *g* for 30 min at 4 °C and subsequently washed using 100µl of 80% acetone, again, centrifuged at 16,000× *g* at 4 °C for 10 min. After drying up at room temperature, 8 M urea with 25 mM ammonium bicarbonate was used to dissolve retinal pellet. Then, 25 mM ammonium bicarbonate was added to dilute 8 M urea to 1 M urea for trypsin digestion. The tryptic digestion of proteins was conducted with an enzyme to protein ratio of 1:25 (*w/w*) at 37 °C for 16 h, and then, the digested peptides were cleaned up by Oasis^®^ HLB 1 cc/10 mg Extraction Cartridge. The eluted peptides were dried up at 4 °C in a vacuum centrifuge, and then the pellet was re-dissolved in 0.1% formic acid (FA) for peptide concentration analysis. Additionally, 0.5µg/ul of digested peptides were stored for LC-MS/MS analysis.

### 4.5. Parameters Setting of IDA and SWATH-MS Experiments

The hybrid quadrupole time-of-flight TripleTOF^®^ 6600 mass spectrometer (SCIEX, Framingham, MA, USA) was selected for information-dependent acquisition (IDA) and SWATH-MS experiments. The retinal digested peptides were loaded onto a trap column (350 µm × 0.5 mm, C18) at 2 µl/min for 15 min by loading buffer (0.1% Formic acid, 2% Acetonitrile in water). It is then separated on a nano-LC column (100 µm × 30 cm, C18, 5 µm) using an Ekisgent 415 nano-LC system (SCIEX, Framingham, MA, USA). LC separation was under 350 nL/min using mobile phase A (0.1% Formic acid, 2% Acetonitrile in water) and B (0.1% formic acid, 98% Acetonitrile in water) with the following gradient: 0–0.5 min: 5%B, 0.5–90 min:10%B, 90–120 min:20%B, 120–130 min: 28%B, 120–135 min: 45%B, 135–141 min: 80%B, 141–155 min: 5%. 10 µm SilicaTip electrospray emitters (New Objective, MA, USA) were used for peptide injection.

For information-dependent acquisition (IDA) experiments, 2.5 µg tryptic peptides from 10 pooled samples were used to build up a proteome library. A total of two technical replicates were included in the IDA experiment to increase protein coverage. IDA scans were performed a high resolution for 250 ms in TOF MS from 350 to 1800 m/z, and then, MS/MS scans were acquired from 100 to 1800 m/z using an accumulation time of 50 ms and with the number of product ion at 50. The criteria for product ion included the intensity of greater than 125 counts/s, charge state from +2 to +5 and mass tolerance of 50 ppm and combined with not included a dynamic exclusion list. For SWATH acquisitions experiments, 2µg of peptide from each individual sample was injected into LC-MS/MS. Overall, three technical replicates were included in SWATH experiment. The 50-ms survey scan (TOF-MS) was performed first and then an isolation of 100 variable windows was selected in a looped mode over the full mass range of 100 to 1800 m/z MS/MS scans. The total cycle time was 3 s. The 100 variable window (VW100) acquisition method was calculated by SCIEX SWATH variable window calculator v.1.0 based on IDA raw data. This calculation method considered the number of precursors and their intensities together as successfully used previously [132].

### 4.6. Protein Ion Library Generation

The 2.5µg tryptic peptides from 10 pooled samples were used to build up a proteome library. The raw data were extracted by the Pro Group™ Algorithms using ProteinPilot 5.0 (SCIEX, Framingham, MA, USA) and searched against the UniProt guinea pig protein database (*Cavia porcellus* (Guinea pig) (10141)) [133]. Iodoacetamide (IAA) was selected as cystein alkylation type, along with trypsin digestion. Biological modification was selected and false discovery rate analysis was also performed. The output is a group file and used for the protein ion library. For the protein identification, the cutoff point is global 1% FDR.

### 4.7. SWATH Processing and Quantitative Analysis

All the files from SWATH experiment were processed using PeakView 2.2 (SCIEX, Framingham, MA, USA). Ion library generated from IDA above was used to support SWATH analysis. A total of 3,800 maximum proteins were imported with unlabeled sample types. The retention time (RT) of all 30 runs was aligned by 10 selected peptides manually covering 20–150 min. SWATH processing settings parameters were: 30 peptides, 6 transitions, peptide confidence of >90%, 1% false discovery rate, exclude shared peptides, and XIC width set at 75 ppm combined with 10 min as extraction time.

After processing, all SWATH files were exported and saved as two kinds of files: (1) Total area files were saved to support quantitative analysis for the next step; (2) All the information was used to support coefficient of variation (CV) analysis using SWATH^®^ Replicates Analysis 2.0 Toolkit (SCIEX, Framingham, MA, USA). Then, all the DIA data in terms of protein level were normalized using total area using MarkerView 1.3 (SCIEX, Framingham, MA, USA). The fold change of differential proteins was expressed by mean of fold change of all biological samples. The *p* value of each protein was measured by paired *t*-test. The criteria for defining differential expressed proteins included (a) the log ratios of proteins were >0.26 for up-regulated proteins or <−0.26 to define down-regulated proteins. (b) *p* value is no more than 0.05 after paired *t*-test.

### 4.8. Bioinformatics Analysis

Identified proteins were matched to gene name by conversion tool in the UniProt protein online database (http://www.uniprot.org/ 15 September 2017). Functional analysis of gene ontology (GO) annotations (biological processes, molecular functions, and cellular components) of all identified retinal proteins was performed using PANTHER gene classification analysis (ver. 12.0, 15 Sep. 2017) [134]. Interacted analysis for differentially expressed proteins was preceded using the search tool for the retrieval of interacting genes and proteins (STRING) v10.5 (15 Sep. 2017) online database [55,135]. Ingenuity pathway analysis (IPA, Ingenuity Systems, Mountain View, CA, USA) was performed with proteins which had significant change in expression in myopic vs. control eyes (*p* ≤ 0.05) [136,137]. Schematic workflow of quantitative discovery proteomics in myopic vs. control eyes was shown in Figure 6.

### 4.9. Protein Validation Using MRM Based Proteomic Approach

To establish the MRM acquisition method, proteins of interest were selected based on the untargeted SWATH datasets. The targeted peptides were firstly checked and extracted from the previously acquired MS spectral IDA library identified by the ProteinPilot. The whole sequence of targeted proteins in FASTA format was downloaded from the UniProt and imported to the Skyline software (MacCoss Lab, Seattle, WA, USA). To fulfill the MRM strategy, the peptides should fulfill the following criteria: (a) the amino acid must be a unique peptide sequence; (b) 3–5 peptides without post-translational modifications; (c) no peptide miss-cleavage with trypsin; (d) the length of peptides was from 8–25 amino acids; (e) the top three highest intensity product ions were selected, and both b and y ions were included; (f) charges of 2+ and 3+ were included with a preference of 2+ for precursor selection; and (g) peptide transitions with retention linear regression of r >0.95 and similarity score of dotp >0.9.

The digested peptides (2µg) from an individual sample (*n* = 5) were loaded onto Halo Peptide-ES C18 column (2.7 µm 160 Å 5 cm) for peptide trapping and Halo Peptide-ES C18 column (2.7 µm 160 Å 5 cm) for peptides separation using M3 MicroLC system (SCIEX, Framingham, MA, USA). The sample was separated at a 15 µL/min flow rate according to the following gradients: in 5% solvent B (98% acetonitrile/0.1% formic acid) during 0.0–1.0 min, then in 5–35% ramping solvent B during 2.0–10.0 min, in 35–50% ramping solvent B during 11.0–12.0 min, in 50–90% ramping solvent B during 13.0–17.0 min, and reducing the solvent B from 90% to 5% during 17.0–17.2 min and keeping in the 5% solvent B for 2.8 min. Solvent A was 95% water/0.1% formic acid. All digested peptides for the MRM analysis were acquired using a QTRAP 6500+ instrument (SCIEX, Framingham, MA, USA) fitted with an electrospray ionization source operated in positive-ion mode, according to the typical parameters as follows: curtain gas (CUR), 20; collision gas (CAD), medium; spray voltage (IS), 5500 V; temperature (TEM), 350 °C; ion-source gas 1 (GS1), 25; and ion-source gas 2 (GS2), 25. The entrance potential (EP) and the collision cell exit potential (CXP) were defaulted to 10 and 7, respectively. The declustering potential (DP) was calculated by linear regression: DP = 0.0729 × (*m/z*) + 31.117. The collision energy (CE) linear regression for doubly charged peptide was calculated using the equation of CE = 0.036 × (*m/z*) + 8.857, and the triply charged peptide was calculated using the equation of CE = 0.0544 × (*m/z*) − 2.4099. The dwell time was 8 msec.

After all the individual biological samples were acquired by the QTRAP 6500+, the MRM raw data were further processed MultiQuant^TM^ (version 3.03, SCIEX, Framingham, MA, USA). The MQ4 algorithm for automatic peak integration, resulting retention time, peak area, peak height, and signal to noise ratio for each monitored transition. The weighted average analysis was performed as in a previous publication [138]. Transition response was calculated from the average of three technical replicates of individual biological sample and type (treated, control). In addition, the transition weighted average was calculated, respectively, with the weight =0 for S/N ratio < 5 and weight = 1 for S/N ratio ≥ 20. A Sigmoid distribution was applied to the intermediate S/N value. The peptide weighted fold-change was calculated as the weighted average of its transitions. The protein weighted fold-change was further calculated as a weighted average of the corresponding individual peptides. The reported standard deviation was calculated from the protein weighted average fold-change of individual biological sample. To improve the accuracy and achieve high signal to noise (S/N) ratio, additional criteria were used to quantify the expressed protein ratios: (a) at least one peptide was selected to calculate the fold changes of one targeted protein; (b) at least two transitions were chosen to calculate the fold change(s) of targeted peptide(s); and (c) any overlapping, saturated, and signals with S/N ratio <5 in transition level were excluded manually.

To date, there is still no standard protocol for interpreting MRM results associated with SWATH. Hence, combining knowledge from the literature and our MS experience, three levels were built to characterize the measured MRM results with increasing confidence: Level 0—represented the lowest confidence, in which inconsistent directional change of protein abundance were found compared treated group to control group between SWATH and MRM; Level 1—indicated a consistent directional change of protein abundance between SWATH and MRM; Level 2—indicated the consistent directional change of protein abundance between SWATH and MRM, the significant *p* value was also needed in the MRM results (*p* ≤ 0.05, paired *t*-test). Hence, proteins meeting Level 2 criteria were considered to be the most validated targets warranting further research.

For the internal standard, the top three high abundant peptides of Glyceraldehyde 3-phosphate dehydrogenase (*GAPDH*) were analyzed in MRM experiments (Appendix A). If there was no alternation of internal standard (*GAPDH*) after MRM experiment, the top one transition was selected with the largest peak area and considered as an internal standard transition. Then each transition peak area was normalized by a matched transition of *GAPDH* in each biological sample. Schematic workflow of quantitative targeted proteomics in myopic vs. control eyes was shown in Figure 7.

### 4.10. Statistical Analysis

The ocular biometric measurements were compared between the myopic and the control eyes in the same animal (two-tailed paired *t*-test) using Microsoft Excel (2007 version). Significance was defined as *p* < 0.05. For protein identification, to minimize false positive results, only proteins at 1% global FDR and non-redundant proteins were included in ion library to support quantitative analysis. To quantify the differential expressed proteins, the three criteria were applied. (a) At least 2 peptides to support protein quantitation. (b) The protein log 2 ratios were larger than 0.26 for upregulated proteins or smaller than minus 0.26 for the downregulated proteins. (c) *p* ≤ 0.05 was determined after paired *t*-test. For protein validation, a statistically significant difference in each quantified proteins between the myopic eyes and the control eyes was also determined by paired two-tailed student’s *t*-test (*p* ≤ 0.05).

## 5. Conclusions

This study showed that the SWATH-based label-free proteomics platform was sensitive to detect minor changes in protein abundance at the early stage of myopia. In addition, the largest number of guinea pig retina proteome reported to date covering early myopia was established. Conventional approaches to investigate biochemical cascades majored in one or a few targets at a time. However, through open-target SWATH-based label-free proteomics approach, for the first time, around 60 up-regulated and down-regulated retinal proteins were successfully reported in one experiment, which has provided system-wide perspective that was difficult to achieve in previous myopia study. By using a series of bioinformatics tool, 90 significant pathways and complex network-associated pathways were also revealed. A vital role of the ‘phototransduction pathway’ was found, with more novel targets uncovered that are associated with this well-known pathway. Moreover, a vital role of lipid metabolism also has been demonstrated by the integration of SWATH-MS and MRM-MS approaches whereby multiple proteins could be confirmed independently at peptide levels without antibodies. In summary, MRM-based targeted proteomics technique was found as an orthogonal approach for validating expressions of multiple proteins in a single MS run. Moreover, hypothesis-free characterization of protein regulation using SWATH-based proteomics combined with targeted validation of protein regulation using MRM identified novel proteins involved in the lipid metabolism at the early stage of myopia development. Our established integrated platform can also be applied to study other retinal diseases.

## Figures and Tables

**Figure 1 ijms-22-04721-f001:**
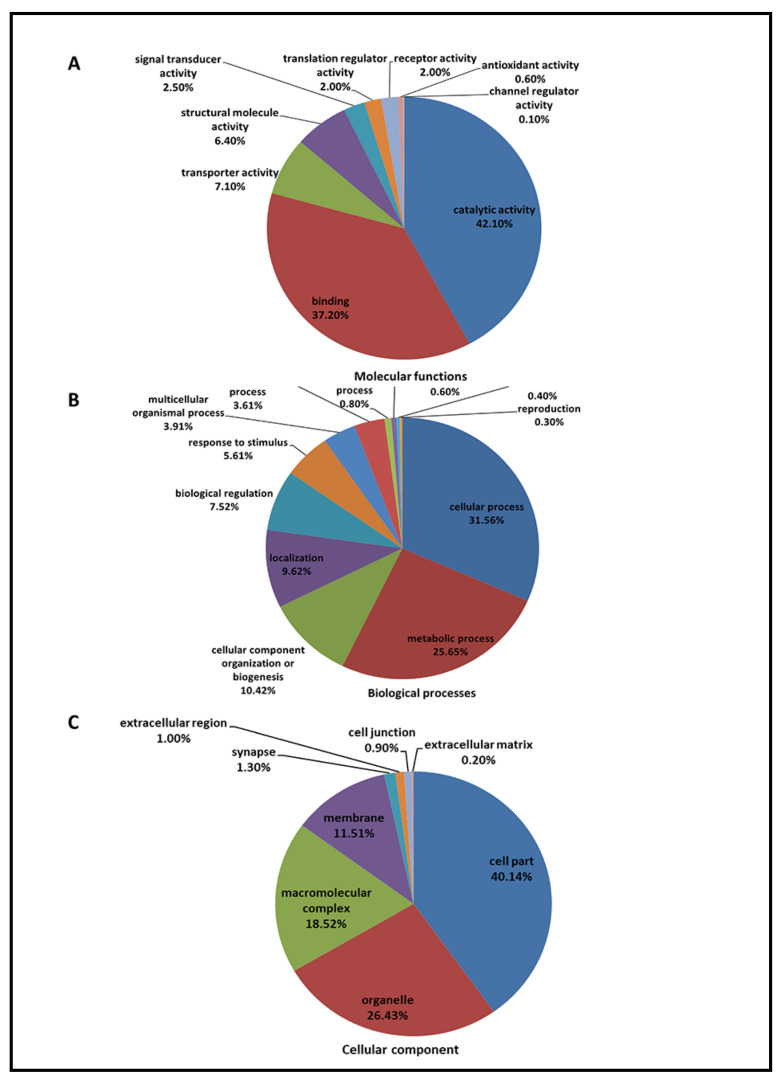
A total of 3202 identified proteins were annotated with the PANTHER^TM^ Classification System (www.pantherdb.org, 15 September 2017) according to their molecular function (**A**), biological process (**B**), and cellular component (**C**).

**Figure 2 ijms-22-04721-f002:**
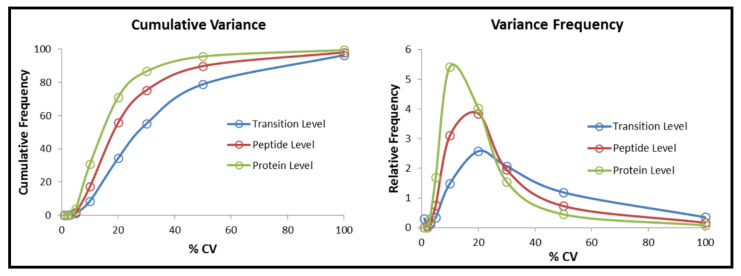
The coefficient of variation (CV) of matched proteins, peptides, and transitions at 100 variable windows size in SWATH acquisition to evaluate the reproducibility and variability of protein quantitation for all 10 biological samples after 4-day LIM.

**Figure 3 ijms-22-04721-f003:**
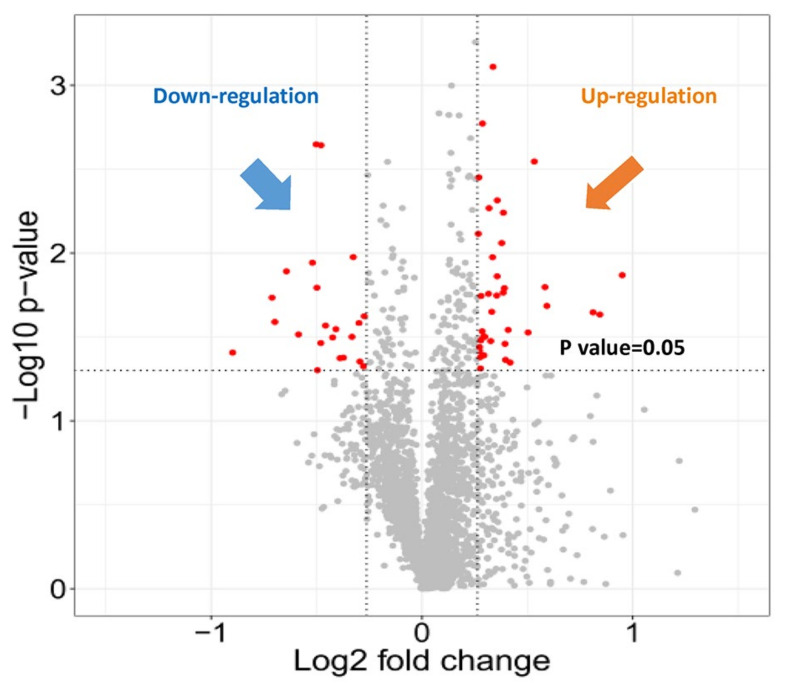
Volcano plot of 58 differentially expressed retinal proteins in myopic eyes after 4-day LIM compared to the corresponding control eyes. Criteria for significant differential expression were as log2 ratio (T/C) >0.26 or <−0.26; *p* ≤ 0.05, paired *t*-test.

**Figure 4 ijms-22-04721-f004:**
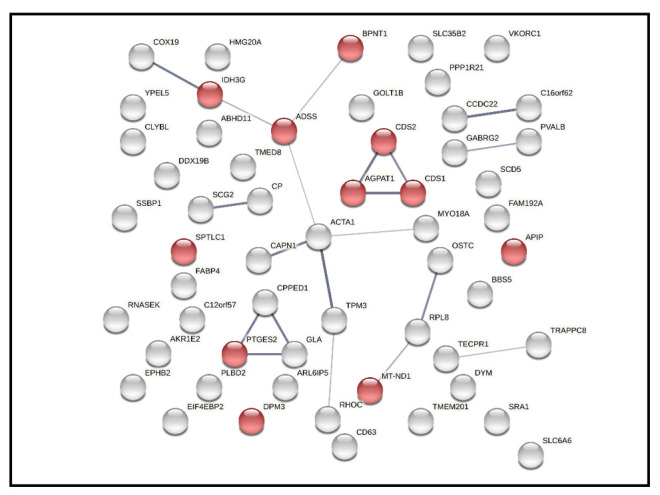
Network analysis of 58 differentially expressed retinal proteins, including 56 nodes (gene names) and 20 edges (predicted functional associations). The closely related proteins are linked by lines, while red color represent proteins associated with metabolic pathways.

**Figure 5 ijms-22-04721-f005:**
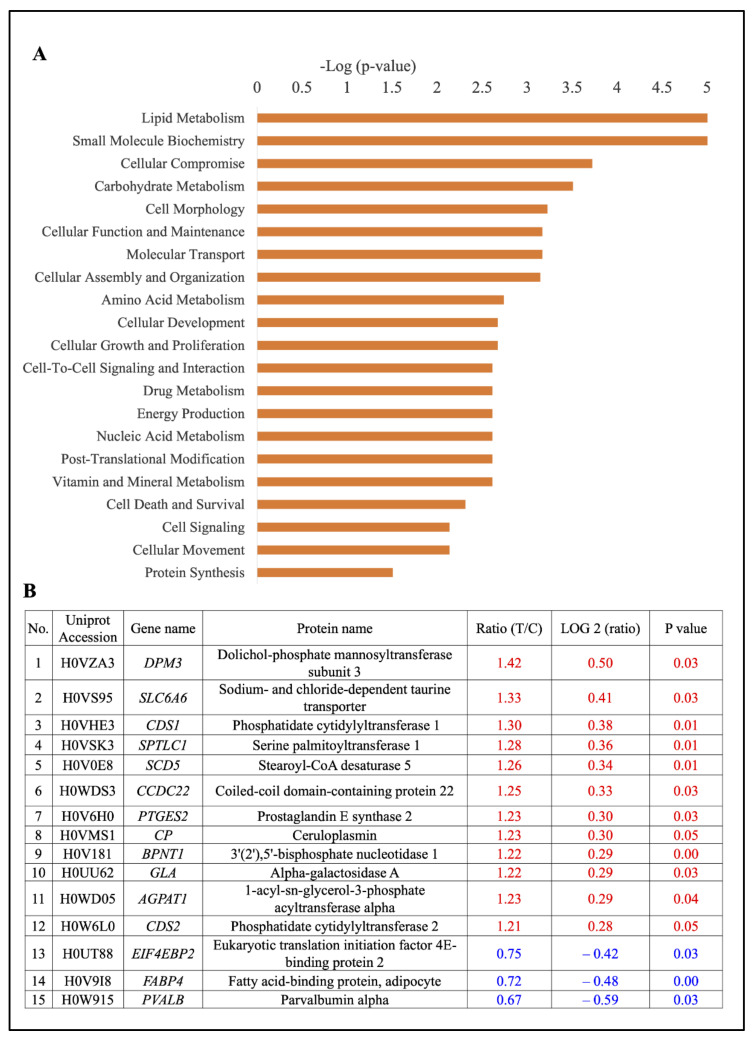
(**A**) Molecular and cellular functions (per GO analysis) significantly associated with the 58 differentially expressed proteins. (**B**) Differentially expressed proteins involved in lipid metabolism. The fold-change (FC), log2 (FC), and *p* values in red belong to up-regulated proteins, and the values in blue belong to down-regulated proteins.

**Figure 6 ijms-22-04721-f006:**
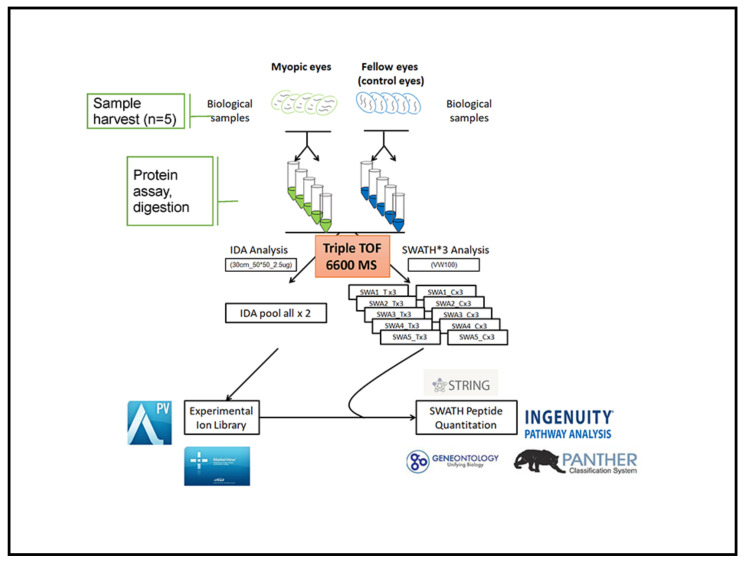
Schematic workflow of quantitative discovery proteomics in myopic vs. control eyes. Overall, 10 retinal lysates from 5 pigmented guinea pigs (five treatment eyes and five control eyes) were included. Approximately 3 µg of tryptic peptide from 10 pooled samples were used in two technical replicates to establish the proteome library under IDA. The 2 µg digested protein from each sample in three technical replicates under SWATH was analyzed by a Triple-TOF 6600 LCMS. Peptide identification from the 2 IDA was consolidated into an ion library for SWATH analysis. Protein identification and retention time calibration for SWATH quantification were performed using the ProteinPilot, PeakView, and MarkerView software, followed by bioinformatics analysis using online tools.

**Figure 7 ijms-22-04721-f007:**
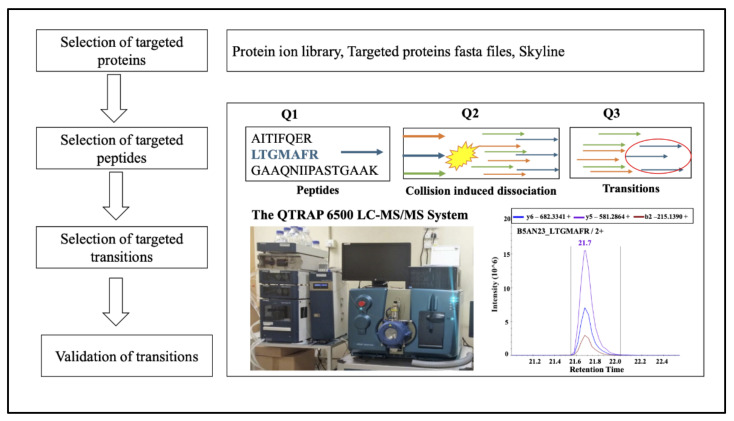
Schematic workflow of quantitative MRM-based targeted proteomics in myopic vs. control eyes (*n* = 5). To establish the MRM acquisition method, proteins of interest were selected based on the untargeted SWATH datasets. The targeted peptides were firstly checked and extracted from the previously acquired MS spectral IDA library identified by the ProteinPilot. The whole sequence of targeted proteins in FASTA format was first downloaded from the UniProt and then imported to the Skyline software (MacCoss Lab, Seattle, WA, USA). The top one transition of GAPDH was selected with the largest peak area and considered as an internal standard transition. The digested peptides (2µg) from each retinal sample were injected into QTRAP 6500. The MRM raw data were further processed MultiQuant^TM^. The MQ4 algorithm for automatic peak integration.

**Table 1 ijms-22-04721-t001:** Refractive errors (in Diopter) and ocular parameters (in mm) of 4-day LIM guinea pigs measured by retinoscopy and high-resolution A-scan ultrasonography, respectively (Mean ± SD, *n* = 5 for each group, * *p* < 0.05, ** *p* < 0.01). ACD and VCD denote anterior chamber depth and vitreous chamber depth, respectively.

Ocular Parameters	Myopic Eyes	Control Eyes	*p* Value
Refractive errors (D)	+5.25 ± 2.33	+6.94 ± 1.81	0.004 **
ACD (mm)	1.089 ± 0.035	1.106 ± 0.052	0.367
Lens thickness (mm)	3.033 ± 0.040	3.024 ± 0.059	0.499
VCD (mm)	2.979 ± 0.035	2.955 ± 0.024	0.253
Retinal (mm)	0.083 ± 0.002	0.083 ± 0.003	1.000
Choroidal (mm)	0.156 ± 0.013	0.174 ± 0.019	0.016 *****
Axial length (mm)	7.416 ± 0.084	7.397 ± 0.104	0.361
Ocular length (mm)	7.756 ± 0.095	7.757 ± 0.122	0.948

**Table 2 ijms-22-04721-t002:** Significant pathways identified by IPA bioinformatic software after 4 days LIM treatment with the *p* values and associated genes, that were in agreement to previous -omics studies. The *p*-value was calculated with the Fischer’s exact test and z-score represents activation or suppression of the corresponding pathway. NaN represents unpredicted.

No	Pathway Name	*p* Value	Z Score	Molecules	Evidence from Gene Expression	Evidence from Protein Expression
1	Phototransduction Pathway	0.000	NaN	*PRKACB*, *GNB1*, *RGS9BP*, *GRK1*, *PDE6B*, *PRKAR1A*, *RCVRN*	[60] (Tkatchenko et al., 2019) [61] (Tkatchenko et al., 2018)	[45] (Shan et al., 2018)
2	Oxidative Phosphorylation	0.000	NaN	*SDHA*, *ATP5B*, *COX5B*, *COX7A2*, *ATPAF2*, *CYB5A*, *NDUFAB1*, *NDUFB10*	[61] (Tkatchenko et al., 2018) [62] (Giummarra et al., 2018)	N/A
3	Mitochondrial Dysfunction	0.000	NaN	*SDHA*, *ATP5B*, *COX5B*, *COX7A2*, *MAPK9*, *ATPAF2*, *CYB5A*, *NDUFAB1*, *NDUFB10*	[61] (Tkatchenko et al., 2018) [63] (Riddell and Crewther, 2017)	N/A
4	IL-1 Signaling	0.000	NaN	*PRKACB*, *GNB1*, *MAPK9*, *GNG13*, *GNG3*, *PRKAR1A*	[60] (Tkatchenko et al., 2019)	N/A
5	Huntington’s Disease Signaling	0.000	NaN	*GNB1*, *SDHA*, *ATP5B*, *MAPK9*, *GNG13*, *DLG4*, *GNG3*, *AP2A2*, *SNAP25*	[60] (Tkatchenko et al., 2019) [62] (Giummarra et al., 2018)	N/A
6	Tight Junction Signaling	0.001	NaN	*PRKACB*, *EPB41*, *TJP2*, *PPP2R2A*, *SNAP25*, *ACTA1*, *PRKAR1A*	[60] (Tkatchenko et al., 2019) [64] (Zhang et al., 2010)	N/A
7	Tec Kinase Signaling	0.001	1.000	*GNB1*, *RHOC*, *PAK2*, *MAPK9*, *GNG13*, *GNG3*, *ACTA1*	[60] (Tkatchenko et al., 2019)	N/A
8	Germ Cell-Sertoli Cell Junction Signaling	0.001	NaN	*EPN1*, *RHOC*, *PAK2*, *ILK*, *MAPK9*, *ACTN4*, *ACTA1*	[60] (Tkatchenko et al., 2019)	N/A
9	Calcium Signaling	0.001	1.000	*PRKACB*, *CAMK2A*, *CAMK2D*, *ATP2B3*, *TPM3*, *ACTA1*, *PRKAR1A*	[60] (Tkatchenko et al., 2019) [61] (Tkatchenko et al., 2018)	N/A
10	α-Adrenergic Signaling	0.001	NaN	*PRKACB*, *GNB1*, *GNG13*, *GNG3*, *PRKAR1A*	[60] (Tkatchenko et al., 2019) [61] (Tkatchenko et al., 2018)	N/A
11	G Beta Gamma Signaling	0.001	0.447	*PRKACB*, *GNB1*, *GNG13*, *GNG3*, *PRKAR1A*	[60] (Tkatchenko et al., 2019)	N/A
12	TCA Cycle II (Eukaryotic)	0.001	NaN	*SDHA*, *IDH3G*, *DLST*	[62] (Giummarra et al., 2018)	N/A
13	CREB Signaling in Neurons	0.001	1.000	*PRKACB*, *GNB1*, *CAMK2A*, *CAMK2D*, *GNG13*, *GNG3*, *PRKAR1A*	[60] (Tkatchenko et al., 2019) [61] (Tkatchenko et al., 2018)	N/A
14	CDP-diacylglycerol Biosynthesis I	0.001	NaN	*CDS1*, *AGPAT1*, *LPGAT1*	[60] (Tkatchenko et al., 2019)	N/A
15	Phosphatidylglycerol Biosynthesis II (Non-plastidic)	0.001	NaN	*CDS1*, *AGPAT1*, *LPGAT1*	[60] (Tkatchenko et al., 2019)	N/A
16	Relaxin Signaling	0.002	NaN	*PRKACB*, *GNB1*, *GNG13*, *GNG3*, *PDE6B*, *PRKAR1A*	[60] (Tkatchenko et al., 2019)	N/A
17	Gαs Signaling	0.003	NaN	*PRKACB*, *GNB1*, *GNG13*, *GNG3*, *PRKAR1A*	[60] (Tkatchenko et al., 2019) [61] (Tkatchenko et al., 2018)	N/A
18	G Protein Signaling Mediated by Tubby	0.003	NaN	*GNB1*, *GNG13*, *GNG3*	[60] (Tkatchenko et al., 2019)	N/A
19	Androgen Signaling	0.003	NaN	*PRKACB*, *GNB1*, *GNG13*, *GNG3*, *PRKAR1A*	[61] (Tkatchenko et al., 2018)	N/A
20	CXCR4 Signaling	0.003	1.000	*GNB1*, *RHOC*, *PAK2*, *MAPK9*, *GNG13*, *GNG3*	[60] (Tkatchenko et al., 2019)	N/A
21	Melatonin Signaling	0.003	−1.000	*PRKACB*, *CAMK2A*, *CAMK2D*, *PRKAR1A*	[60] (Tkatchenko et al., 2019)	[65] (Rada and Wiechmann, 2006)
22	Ephrin B Signaling	0.004	NaN	*GNB1*, *EPHB2*, *GNG13*, *GNG3*	[61] (Tkatchenko et al., 2018)	N/A
23	Gαi Signaling	0.004	NaN	*PRKACB*, *GNB1*, *GNG13*, *GNG3*, *PRKAR1A*	[60] (Tkatchenko et al., 2019) [66] (Srinivasalu et al., 2018)	N/A
24	Purine Nucleotides De Novo Biosynthesis II	0.004	NaN	*ADSS*, *PAICS*	[60] (Tkatchenko et al., 2019)	N/A
25	RhoGDI Signaling	0.004	NaN	*GNB1*, *RHOC*, *PAK2*, *GNG13*, *GNG3*, *ACTA1*	[60] (Tkatchenko et al., 2019)	N/A
26	Gustation Pathway	0.004	NaN	*PRKACB*, *GNB1*, *GNG13*, *PDE6B*, *PRKAR1A*	[61] (Tkatchenko et al., 2018)	N/A
27	Oleate Biosynthesis II (Animals)	0.005	NaN	*SCD5*, *CYB5A*	[60] (Tkatchenko et al., 2019)	N/A
28	Signaling by Rho Family GTPases	0.005	1.000	*GNB1*, *RHOC*, *PAK2*, *MAPK9*, *GNG13*, *GNG3*, *ACTA1*	[60] (Tkatchenko et al., 2019)	N/A
29	P2Y Purigenic Receptor Signaling Pathway	0.006	NaN	*PRKACB*, *GNB1*, *GNG13*, *GNG3*, *PRKAR1A*	[60] (Tkatchenko et al., 2019)	N/A
30	Protein Kinase A Signaling	0.007	1.633	*PRKACB*, *GNB1*, *CAMK2A*, *CAMK2D*, *GNG13*, *GNG3*, *GRK1*, *PDE6B*, *PRKAR1A*	[60] (Tkatchenko et al., 2019)[61] (Tkatchenko et al., 2018)	N/A
31	ILK Signaling	0.007	0.816	*PPP2R2A*, *RHOC*, *ILK*, *MAPK9*, *ACTN4*, *ACTA1*	[60] (Tkatchenko et al., 2019)[67] (Wu et al., 2018)	N/A
32	IL-8 Signaling	0.007	1.000	*GNB1*, *RHOC*, *PAK2*, *MAPK9*, *GNG13*, *GNG3*	[60] (Tkatchenko et al., 2019)	N/A
33	Clathrin-mediated Endocytosis Signaling	0.007	NaN	*EPN1*, *SYNJ1*, *RAB11B*, *EPHB2*, *AP2A2*, *ACTA1*	[60] (Tkatchenko et al., 2019)	N/A
34	D-myo-inositol (1,4,5)-trisphosphate Degradation	0.010	NaN	*SYNJ1*, *BPNT1*	[60] (Tkatchenko et al., 2019)	N/A
35	CDK5 Signaling	0.010	−1.000	*PRKACB*, *PPP2R2A*, *MAPK9*, *PRKAR1A*	[60] (Tkatchenko et al., 2019)	N/A
36	Regulation of eIF4 and p70S6K Signaling	0.011	NaN	*RPS7*, *EIF4EBP2*, *PPP2R2A*, *EIF2B2*, *EIF3M*	[60] (Tkatchenko et al., 2019)[61] (Tkatchenko et al., 2018) [68] (Zelinka et al., 2016)	N/A
37	EIF2 Signaling	0.012	NaN	*RPS7*, *RPL8*, *EIF2B2*, *ACTA1*, *RPLP0*, *EIF3M*	[60] (Tkatchenko et al., 2019) [61] (Tkatchenko et al., 2018) [ [69](Yang et al., 2015) [67] (Wu et al., 2018)	N/A
38	Spermine Biosynthesis	0.017	NaN	*SMS*	[61] (Tkatchenko et al., 2018)	
39	Ephrin Receptor Signaling	0.017	NaN	*GNB1*, *EPHB2*, *PAK2*, *GNG13*, *GNG3*	[60] (Tkatchenko et al., 2019)[61] (Tkatchenko et al., 2018)	N/A
40	Super-pathway of D-myo-inositol (1,4,5)-trisphosphate Metabolism	0.018	NaN	*SYNJ1*, *BPNT1*	[60] (Tkatchenko et al., 2019)	N/A
41	Synaptic Long Term Potentiation	0.020	1.000	*PRKACB*, *CAMK2A*, *CAMK2D*, *PRKAR1A*	[60] (Tkatchenko et al., 2019) [61] (Tkatchenko et al., 2018)	N/A
42	RAR Activation	0.024	NaN	*PRKACB*, *SRA1*, *MAPK9*, *CRABP1*, *PRKAR1A*	[60] (Tkatchenko et al., 2019) [70] (Huang et al., 2011)	[70] (Huang et al., 2011) [71] (Huo et al., 2013)
43	S-methyl-5-thio-α-D-ribose 1-phosphate Degradation	0.026	NaN	*APIP*	[61] (Tkatchenko et al., 2018)	N/A
44	Inosine-5’-phosphate Biosynthesis II	0.026	NaN	*PAICS*	[60] (Tkatchenko et al., 2019)	N/A
45	NRF2-mediated Oxidative Stress Response	0.026	NaN	*AKR1A1*, *MAPK9*, *DNAJA1*, *MGST3*, *ACTA1*	[60] (Tkatchenko et al., 2019) [67] (Wu et al., 2018)	N/A
46	Antiproliferative Role of Somatostatin Receptor 2	0.027	NaN	*GNB1*, *GNG13*, *GNG3*	[60] (Tkatchenko et al., 2019)[61] (Tkatchenko et al., 2018)	N/A
47	BMP signaling pathway	0.028	NaN	*PRKACB*, *MAPK9*, *PRKAR1A*	[72] (Li et al., 2015)	[73] (Li et al., 2016)
48	Dopamine Receptor Signaling	0.028	NaN	*PRKACB*, *PPP2R2A*, *PRKAR1A*	[60] (Tkatchenko et al., 2019) [61] (Tkatchenko et al., 2018)	[74] (Feldkaemper and Schaeffel, 2013) [75] (Chen et al., 2017) [76] (Zhou et al., 2017)
49	Insulin Receptor Signaling	0.033	0.000	*PRKACB*, *SYNJ1*, *EIF2B2*, *PRKAR1A*	[60] (Tkatchenko et al., 2019) [77] (He et al., 2014) [78] (Penha et al., 2011)	[79] (Ritchey et al., 2012)
50	Xenobiotic Metabolism Signaling	0.037	NaN	*SRA1*, *CAMK2A*, *CAMK2D*, *PPP2R2A*, *MAPK9*, *MGST3*	[60] (Tkatchenko et al., 2019)	N/A
51	HIPPO signaling	0.039	NaN	*TJP2*, *PPP2R2A*, *DLG4*	[60] (Tkatchenko et al., 2019) [61] (Tkatchenko et al., 2018)	N/A
52	Integrin Signaling	0.041	−0.447	*RHOC*, *PAK2*, *ILK*, *ACTN4*, *ACTA1*	[60] (Tkatchenko et al., 2019)[61] (Tkatchenko et al., 2018) [80] (Tian et al., 2013)	N/A
53	cAMP-mediated signaling	0.044	0.447	*PRKACB*, *CAMK2A*, *CAMK2D*, *PDE6B*, *PRKAR1A*	[61] (Tkatchenko et al., 2018) [81] (Tao et al., 2013)	[82](Chun et al., 2015)
54	Regulation of Actin-based Motility by Rho	0.044	NaN	*RHOC*, *PAK2*, *ACTA1*	[60] (Tkatchenko et al., 2019) [40] (Wu et al., 2018a)	N/A
55	Netrin Signaling	0.044	NaN	*PRKACB*, *PRKAR1A*	[60] (Tkatchenko et al., 2019)	N/A
56	Gαq Signaling	0.049	NaN	*GNB1*, *RHOC*, *GNG13*, *GNG3*	[60] (Tkatchenko et al., 2019)	N/A

**Table 3 ijms-22-04721-t003:** The validation results of 7 differentially expressed proteins involved in lipid metabolism by MRM based proteomic approach (*n* = 5 for each group, * *p* < 0.05).

No	UniProt Accession	Gene Name	Protein Description	SWATH		MRM		
				Protein FC (T/C)	*p* Value	Protein FC (T/C)	*p* Value	Confidence
1	H0VS95	*SLC6A6*	Sodium- and chloride-dependent taurine transporter	1.33	0.030	1.16	0.039 *	2
2	A0A286XMC0	*PTGES2*	Prostaglandin E synthase 2	1.23	0.030	1.13	0.040 *	2
3	A0A286XCE4	*CP*	Ceruloplasmin	1.23	0.050	1.12	0.527	1
4	A0A286XGK4	*CDS1*	Phosphatidate cytidylyltransferase 1	1.30	0.010	1.31	0.083	1
5	H0VSK3	*SPTLC1*	Serine palmitoyltransferase 1	1.28	0.010	1.05	0.352	1
6	H0WDS3	*CCDC22*	Coiled-coil domain-containing protein 22	1.25	0.030	1.12	0.350	1
7	H0UU62	*GLA*	Alpha-galactosidase A	1.22	0.030	1.08	0.356	1

## Data Availability

The data presented in this study are available in request from the corresponding author.

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
