# Peer review of "SWATH Based Quantitative Proteomics Reveals Significant Lipid Metabolism in Early Myopic Guinea Pig Retina"

_ijms, 2021, doi:10.3390/ijms22094721_

Round 1

Reviewer 1 Report

General comments to authors

Bian etal., provides a comprehensive analysis of the protein expression profile using a model of myopia, to give a better understanding of the pathophysiology of this condition. The article is properly written, the methods are acceptable, and the results are properly presented. I however have raised a few queries that need to eb addressed.

Please revise the article for minor grammatical errors, and for scientific accuracy. What is the purpose of the following statement ‘’Error! Reference source not found’’ which is appearing in many places of the text throughout the article, for example page 6 line 190.

Specific comments for authors

  1. On page 4, lines 137 to 139, authors write that ‘’the top 2000 proteins in the retinal ion library were uploaded to STRING to check interactions among them’’, but the question is how where these proteins filtered for, and why the top 2000 and not 1000, or even 3000, please justify this?
  2. In figure 1, there is no meaningful information that be taken by just looking at the figure. It is too congested and as such impossible to make anything out of it. Please consider moving the current figure into the supplementary and may be generate a new zoomed in figure from the same illustrating a close up of a few meaningful protein-protein interactions. In addition, I suggest that functional based interaction analysis be done, which is more relevant than the currently presented data. Please consider revising this analysis to show functional annotation (i.e., To illustrate how the genes interact biologically with each other, this can be done very simply in STRING) as biological interactions have more meaning in terms of potential follow up experimental studies.
  3. Page 4, lines 148 to 157, functional annotation analysis can be done in STRING as well, please explain the reason as to why PANTHER was used for the functional annotation analysis? Of what relevance is the functional annotation analysis results. i.e., what does it mean to have ‘’molecular functions of retinal proteins “catalytic activity” (42.10%), “binding” (37.20%), and “transporter activity” (7.10%), because any list of genes feed into PANTHER will always yield similar results.
  4. On page 6 lines 195 to 196, authors write that: ‘’The top one significant pathway was phototransduction pathway’’ however, this pathway is neither active nor suppressed in terms of the zscore, then what makes this pathway interesting? Also, why are pathways such as Sertoli Cell-Sertoli Cell Junction Signaling, Cardiac β-adrenergic Signaling, and Cardiac Hypertrophy Signaling among others of interest if they have neither a z score, and nor evidence from both gene and from protein expression?
  5. Page 7. Food for thought! Are all the pathways listed in Table 2 relevant for myopia? If not justify their inclusion in this table, otherwise limit this table to only the relevant pathways.
  6. Authors write about ‘’Quantitative analysis of differentially expressed proteins’’ however, it is not explained how the differentially expressed proteins were obtained in the first place? What was the cut off fold change used?
  7. Methods page 7 lines 471 to 472, the word is ‘’ad libitum’’ and not ‘’at libitum.’’
  8. Consider including a separate subheading titled ‘statistics’ and under this describe the statistical consideration made in the study. For example, define the p value used, cut off fold change, statistical tests used and why, statistical programs used for the data analysis, format I which the data presented in the manuscript etc.

Reviewer 2 Report

   This is an interesting and well-made work heavily based on proteomics, in which the differential proteomes of retina along myopic development are analysed. Two main quantitative targeted proteomic approaches are used with such purpose, SWATH and MRM that, among others, allowed the characterization of the retinal proteome, of the differentially expressed proteins and related biochemical pathways in the early stage of the myopia. In particular 37 upregulated and 21 downregulated proteins were identified, with validation of index proteins associated with the lipid metabolism.

   The works appears as sound, well done, following the presently accepted standards, including the statistics requirements, and positively contributing to its field. Among suggestions for improvement it can be indicated that:

-The present typescript, in spite that is long (31 pages and 132 references), is quite specialized and a bit heavy for general readers. For the benefit and increased interest of the latter, authors should make an effort to clarify in the text, in plain words, the main fundamental technical aspects of it, like the key related and differential aspects of SWATH and MRM, the information they generate, and the benefits of its combined use, in general and in this particular work.

-The typescript have to be revised for typing mistakes like the ones (at least in the reviewer's copy) that appear in lines 110-111, 113, 116, 119 ...etc, probably related with the automated connection between reference numbers in the main text and the list of references at the end of the paper. Also, for the lack of authors names at the beginning of references 23 and 46, in the list.
